# Identification of Individual *Hanwoo* Cattle by Muzzle Pattern Images through Deep Learning

**DOI:** 10.3390/ani13182856

**Published:** 2023-09-08

**Authors:** Taejun Lee, Youngjun Na, Beob Gyun Kim, Sangrak Lee, Yongjun Choi

**Affiliations:** 1Department of Animal Science, Konkuk University, Seoul 05029, Republic of Korea; tjlee96@naver.com (T.L.); ruminoreticulum@gmail.com (Y.N.); bgkim@konkuk.ac.kr (B.G.K.); leesr@konkuk.ac.kr (S.L.); 2Animal Data Laboratory, Antller Inc., Seoul 05029, Republic of Korea

**Keywords:** cattle identification, deep learning, transfer learning, Efficientnet, *Hanwoo*, muzzle pattern

## Abstract

**Simple Summary:**

Cattle identification is necessary for precision feeding and management. Cattle muzzles have unique patterns, which can be used as a biometric identification key. This study aimed to identify cattle via a deep learning model based on muzzle images. Muzzle patterns were cropped from images using the YOLO v8-based image cropping model. Various artificial intelligence models based on neural networks were studied through transfer learning cropped images for cattle recognition with four optimizers. Several models showed a high prediction accuracy of over 97 percent, implicating the possibility for real farm usage.

**Abstract:**

The objective of this study was to identify *Hanwoo* cattle via a deep-learning model using muzzle images. A total of 9230 images from 336 *Hanwoo* were used. Images of the same individuals were taken at four different times to avoid overfitted models. Muzzle images were cropped by the YOLO v8-based model trained with 150 images with manual annotation. Data blocks were composed of image and national livestock traceability numbers and were randomly selected and stored as train, validation test data. Transfer learning was performed with the tiny, small and medium versions of Efficientnet v2 models with SGD, RMSProp, Adam and Lion optimizers. The small version using Lion showed the best validation accuracy of 0.981 in 36 epochs within 12 transfer-learned models. The top five models achieved the best validation accuracy and were evaluated with the training data for practical usage. The small version using Adam showed the best test accuracy of 0.970, but the small version using RMSProp showed the lowest repeated error. Results with high accuracy prediction in this study demonstrated the potential of muzzle patterns as an identification key for individual cattle.

## 1. Introduction

*Hanwoo* (Korean native beef cattle, *Bos taurus coreanae*) is the dominant meat cattle species in Korea; *Hanwoo* cattle are reared in restricted-size pastures and provided with cattle feed due to limited land availability for husbandry purposes. Consequently, the meat prices are inevitably higher compared to those of products from overseas, which are produced in more extensive grazing areas. To maintain the competitiveness of *Hanwoo* cattle, the aim is to achieve premium meat production through specific precision breeding. Precision feeding tailored to individual requirements reduces feed wastage, economizes feed expenses and mitigates methane emissions resulting from excessive intake [1]. However, achieving precision specifications demands precise individual recognition methods.

Currently, tagging is the most popular identification method for cattle. Without exception, each individual cattle is managed with a Korean national traceability number containing 12 digits [2]. According to the study of Fosgate, the retention period of ear tags is 272 days on average, which was relatively shorter than expected [3]. Fosgate suggested relying solely on ear tags for long-term identification might not be sufficient, and close resemblance of the appearance among *Hanwoo* cattle might increase the risk of failure in individual recognition due to tag losses (Figure 1). To address the issue of ear-tag losses, employing supplementary identification methods can aid in ensuring recognition even in cases of ear-tag loss. While RFID chips attached to ear tags are a cheap and convenient technology used to identify cattle, a dedicated reader is needed. Furthermore, a Korean slaughter center runs a system in the base of RFID chips, and some errors are made during grading assessments by reading the wrong chip.

Before the research, a probe was set to find supplementary methods utilizing inherent physical characteristics resistant to alteration, replication or counterfeiting, without requiring specialized identification devices. Biometrics are the identification method used for human individuals based on biological characteristics including face traits, iris patterns, and vein images [4]. Biometrics are highly reliable, consistent with their utilization in banks and airports. Within biometric traits, fingerprints have achieved a level of mainstream adoption, such as mobile phones and door locks. Cattle have unique muzzle patterns on their nose with similar ridge-valley characteristics and individuality to the human fingerprints providing immunity to alteration, replication, or counterfeiting [5].

Computer vision is a technology used for object detection, image classification and other tasks by analyzing images [6]. Presently, computer vision is one of the most rapidly advancing fields closely intertwined with the realm of artificial intelligence, due to the evolution of computer hardware and the proliferation of imagery accessible through the widespread use of mobile phones equipped with cameras [7]. Machine learning algorithms are algorithm improving automatically through experience from given image data [8]. If machine learning algorithms provide inaccurate predictions, engineers need to intervene and make adjustments. When using deep learning models, the algorithm can assess the accuracy of predictions through its own neural network, eliminating the need for human assistance.

Several studies have been conducted to use muzzle patterns to identify cattle and have provided results with various species of cattle [9,10]. *Hanwoo* cattle have no pattern or color differences on the muzzle, but previous research did not investigate breeds characterized by monochromaticity. Therefore, the objective of this study was to identify individual *Hanwoo* cattle through images of muzzle patterns through deep learning.

## 2. Materials and Methods

### 2.1. Training Dataset 

A total of 9230 muzzle images from 336 individual *Hanwoo* cattle were constructed from two beef farms (Table 1). No direct contact was made while collecting the image data, and images were collected using cell phone cameras (iPhone Xs, Apple, Cupertino, CA, USA, and Galaxy s 20, Samsung, Suwon, South Korea) from 24 August 2021 to 12 November 2021 (Figure 2).

The distance between cattle and camera was pinned at 30 cm with the muzzle pattern centered in the image. To capture whole muzzle patterns, capturing whole bottom lips were the essential requirements. For each individual cattle, approximately 20 images were captured, and five pictures were taken four times each to avoid overfitting the model. Some images containing foreign substances were added.

Manual annotations of the muzzle pattern area were added to 150 images using a commercial annotation tool (roboflow; https://roboflow.com (accessed on 17 July 2023)), and the results were saved in the form of bounding box coordinates in xml files. Guidelines for this annotation were: 1. Configure the endline setting to include the whole muzzle area. 2. Include the line of the bottom lip. A model detecting the muzzle pattern area was developed with 150 images and bounding box coordinates using the YOLO v8 algorithm [11]. Consequently, the identified bounding box coordinates of muzzle area were detected for the remaining data. Muzzle areas in images were cropped with xml files using the Python Imaging Library in order to make the dataset used to train the cropping model (Figure 3). The entire image dataset was divided into three categories: training, validation and testing, with split ratios of 75 percent, 15 percent and ten percent, respectively, according to the guidelines of Yadav and Shukla [8]. Training and validation datasets were shown through the model for the transfer learning, and test datasets were not shown when evaluating the practical application.

### 2.2. Data Transformation

Data transformation is useful when modifying data in order to reduce bias [12]. Resize, rotation, flip, warp in frame, adjustment of brightness and contrast can be accomplished in order to decrease the similarity of interference from surrounding factors in the image. Images in each dataset were resized to 224 × 224 pixels, randomly flipped in vertical and horizontal axis and rotated randomly within the range of 20 degrees. After the transformation, image data were normalized by subtracting the mean value for each RGB channel (0.485, 0.456, 0.406 in order) and dividing by standard deviations (0.229, 0.224, 0.225) (Table 2) [13]. The normalization process maintains the transformed image data with a mean close to zero and a standard deviation close to one.

### 2.3. Data Loader

Data blocks with images and labels were combined to make an image-based classifying model. In Korea, each individual cattle is managed with a twelve-digit number code called the livestock traceability number. The livestock traceability number is given at birth and used until slaughter and distribution. Each muzzle-cropped image and matching national livestock traceability numbers were combined in a data block. Data blocks of train and validation sets were sent to each GPU device separately, by batch sizes set to 32. The term ‘epoch’ refers to a time during which the entire dataset undergoes one complete calculation. Training data were used to train the model through the epoch, and validation data were used to validate the model at the end of the epoch.

### 2.4. Transfer Learning

In classification models, a loss serves as a metric indicating the disparity between the model’s predictions and the actual targets, aiming to minimize this discrepancy during the training [14]. The two-by-two confusion matrix is a commonly used method for interpretation, composed of actual negative and positive as rows and predicted negative and positive as columns. By adopting the confusion matrix, all predictions can be sorted into four classes based on the relationship between predicted and actual values (Table 3) [15]. Accuracy mentioned in this paper refers to the sum of true positive and negative divided by the total number of predictions:(1)$$Accuray=\frac{True\;positive+True\;negative}{Total\;number\;of\;prediction}$$

The parameters used to train were model, loss function, optimizer algorithm, learning rate management scheduler and the number of epochs. Each epoch was segmented into the train phase and the validation phase, and recorded the loss and accuracy. At the beginning of each epoch, inputs for each data batch along with labels were received and the gradients of the parameters were initialized through the optimizer. During the training phase, model calculations through forward propagation on the inputs were tracked to generate outputs and passed to the loss function. The loss gradients for each parameter received from the forward propagation were computed by the loss function in the backpropagation stage. Optimizer updated parameters in directions with loss gradients to minimize the loss. If the performance of the epoch’s training results exceeded the previous best during the validation phase, the model’s weights were deemed optimal and saved.

#### 2.4.1. Model

Efficientnet v2 models were opted because of the availability of random weight versions designed for custom datasets [16]. EfficientNet v2 represented an advancement in neural network architecture aiming to enhance efficiency and performance beyond conventional convolutional methods. The essence lies in co-optimizing network depth, width and resolution through compound scaling. Unlike previous models, compound scaling balances dimensions for minimized resource usage and maintained accuracy. EfficientNet v2 refines the original architecture, offering variants tailored to balance computation and performance. Noteworthy features include effective high-resolution image handling, compact model preservation, and a range of variants for trade-offs between size and performance.

The architecture of EfficientNet v2 comprises three key components: the backbone, fused-mobile vision convolutional network (MBConv) layer, and scaling and kernel size adjustment [17]. Backbone of EfficientNet v2 was designed through Neural Architecture Search (NAS) to strike a balance between computation and parameter efficiency, offering a balance between computational complexity and performance, with the ability to scale the model to various sizes [18]. Fused-MBConv layers enhances operational efficiency by combining convolutional layer and depth-wise convolution layer (DWConv) at the early stage from image input [16]. EfficientNet v2 employs a scaling strategy to expand the model’s size while maintaining computational efficiency, and Figure 4 demonstrated the overall schematic of the small model (EfficientNet v2-S) (Figure 4).

Three variations of EfficientNet v2 models (tiny, small and medium) were used. While these three models perform similar roles, they differ in terms of the number of parameters, Giga multiply accumulates (GMACs), and number of activations (Table 4). Parameters in the model refer to the weights and biases for learning. GMACs serves as a metric indicating the number of operations performed by a model, representing its computational complexity [19]. Activations are the total number of intermediate values generated between each layer within the model. Each layer transforms input data through weights and biases, applying the results to activation functions before passing them to the next layer.

#### 2.4.2. Loss Function

The criterion used for the loss function was cross entropy loss applied across all models. Cross entropy loss is commonly used loss function in classification. The difference between the actual class and the predicted class is measured to guide the learning process of the model. Cross entropy loss is calculated as
(2)$$Loss=-\frac{1}{N}\sum_{n=1}^{N}log(\sum_{i}q_{{\tilde{y}}_{n}}i\hat{p}\left(y^{*}=i|x_{n}\right))$$
where *N* is the number of training samples, $\tilde{y}$ represents predicted class, *y** represents the actual class and *x* represents the input image [20].

#### 2.4.3. Optimizer Algorithms 

Stochastic gradient descent (SGD), root-mean-square propagation (RMSProp), adaptive moment estimation (Adam) and evolved sign momentum (Lion) were used as optimizers.

SGD algorithm estimates the gradient on randomly picked batch *z_t_*:(3)$$w_{t+1}=w_{t}-\gamma_{t}\nabla Q(z_{t},w_{t})$$
where w is the weight, γ is learning rate and $Q(z_{t},w_{t})$ is the loss function [21]. With this equation, weights are updated in each direction to minimize the loss.

RMSProp algorithm works similar to SGD but apply different learning rates to each parameter by moving average of the squared gradients, whereas SGD uses a fixed learning rate [22].

Adam algorithm tracks the decaying average of the past gradients, term of momentum, and corrects bias occurred from moving the average of the squared gradients in the RMSProp algorithm [23].

The Lion optimizer is a trending algorithm which is more memory-efficient than the Adam algorithm and with a similar structure; it can be utilized by adjusting the learning rate and weight decay simultaneously [24].

#### 2.4.4. Learning Rate Schedular

The learning rate is a scaled value applied to the gradient to update the weights. A lower learning rate requires more epochs for effective learning, whereas a higher learning rate could potentially hinder the reduction in loss [25]. The learning rate was set to the basic level of 0.001 for all optimizers: SGD, MRS Prop, Adam and Lion. The learning rate was adjusted every seven epochs with a reduction ratio of 0.1, consistently applied across all models [26].

#### 2.4.5. Epochs

Epochs for learning were pinned to 100 for the whole learning process.

### 2.5. Computing Resources

The experiments were conducted on a computing cluster with the following specifications:CPU: Intel(R) Xeon(R) w5-3433 1.99 GHz;RAM: 256 GB of DDR4 RAM;GPU: NVIDIA GeForce RTX 4090.

The deep learning models were implemented using the following software configurations:Operating system: Microsoft Windows 10.0.19045.3324 version 22H2;CUDA: CUDA Version: 11.8;Python: 3.11.4;PyTorch: 2.0.0.

## 3. Results

### 3.1. Loss and Accuracy Metrics through Transfer Learning through Train and Validation Data

Training loss, training accuracy, validation loss and validation accuracy were tracked throughout the entire transfer learning for all models. The quantity of all the metrics is vast since there are 100 epochs from 12 models. The flow of each of the metrics is visualized on each single graph in Figure 5 and Figure 6, and specific numeric metrics of each model from the best validation epoch accuracy are presented in Table 5.

### 3.2. Prediction Result on Test Dataset 

The top five models showing high validation accuracies with a low time taken for each epoch were evaluated through predictions made on test data. Accuracy, testing time, overall number of errors and repeated errors on the same objects of prediction on the test dataset are presented in Table 6.

## 4. Discussion

The deep learning models were investigated for their use in identifying individual cattle based on whole muzzle images at the beginning of the study. However, each image contained an average of 4 MB of information, and the database consisted of 17,325 images, resulting in a large dataset size of approximately 69.3 GB. Processing a significant amount of data led to lengthy analysis and model response times [27]. To make the analysis more practical for real-world farm use, the idea of cropping the muzzle area was experimented with. 

To build a high-quality model with accurate predictions, the selection of an appropriate model is crucial. The artificial neural network converts image data into one dimension, finds characteristics, and then laminates them to highlight the features [28]. Within the models varying in parameter sizes, the small model showed better validation accuracy in comparison to the tiny and medium models for the same optimizer. In terms of optimizer, SGD algorithms were not fit for the Efficientnet V2 model and images in dataset showing very low metrics including validation accuracy. Considering the optimizer as a focal point, overall validation accuracies were higher in Lion and Adam algorithms with similar metrics than RMSProp, but small Efficientnet V2 model with RMSProp showed the best performance in testing practical use.

Despite the successful development of a deep learning model for individual *Hanwoo* cattle identification using muzzle patterns, there are certain limitations in this study. One limitation is the reliance on image data captured through cell phone cameras, which could potentially lead to variations in image quality and consistency. Additionally, the model’s performance might be influenced by changes in angles and other environmental factors during image capture. 

Additionally, the model’s accuracy could be impacted by the challenges of recognizing cattle in real-world scenarios. Factors including changes in cattle appearance due to weather conditions, or health status could affect the model’s reliability in practical farming environments. Furthermore, the transition from research to real-world applications may require addressing issues related to computational resources, deployment and integration with existing farm management systems. Finally, while the achieved predictive accuracy is promising, the model’s performance should be further validated and refined using larger and more diverse datasets. Expanding the dataset to include a wider range of cattle breeds and environmental conditions will help assess the model’s generalization capabilities and identify potential biases.

Inscription and fingerprints share similarities, but there are notable differences in their measurement methods. Fingerprints exhibit one-dimensional characteristics as they are obtained by pressing the finger onto a surface, which captures the unique patterns through pressure [29]. On the other hand, inscriptions are relatively more complex images with three-dimensional characteristics. They require data to be obtained through camera images, capturing the intricate details and depth of the inscribed patterns. The complex architecture should be capable of capturing and processing the detailed depth information inherent in the inscriptions, enabling more precise predictions to be made. It seems that these complexities indeed necessitate a more complex architecture in order to make accurate predictions.

*Hanwoo* cattle have light brown fur with no specific patterns or colors on its face or muzzle (Figure 1), while *Holstein* cattle have black and white fur with a variety of patterns. Furthermore, the muzzle area of *Holstein* cattle can range from pink to black or brown in color. Prior to initiating the current study, the ResNet architecture, commonly utilized in image classification, was used to train models to explore the possibilities of muzzle pattern identification for both *Hanwoo* and *Holstein*. The *Holstein*-identifying model showed better performance than the *Hanwoo*-identifying model. The difference in performance between the *Hanwoo* and *Holstein* models prompted us to consider that the model might have learned more of the colors of the muzzle area or the overall face, rather than solely capturing the unique patterns of ridges and valleys.

While the high accuracy of the *Holstein* model leads to the perception of its strong performance, but if the model relies on colors and skin pattern, colors and patterns might become a hurdle for recognizing ridges and valley patterns of the muzzle across various breeds in the future. Therefore, this study aimed to ensure that the model focuses primarily on the patterns and not just the color variations. After research on the *Hanwoo* breed, an identification model for *Holstein* cattle was developed through the same methods, with 7835 images of 297 *Holstein* cattle. The small Efficientnet V2 model, presented as the best performing model for real usage in this research, showed 0.953 predictive accuracy on test data, presenting subtle variance compared with the *Hanwoo* model, which even showed better performance.

The object of this study was to identify *Hanwoo* cattle using the deep learning model trained with muzzle images, as mentioned in the Introduction. For this study to hold significance, the developed models must be applicable in practical industry. The high predictive accuracy on unseen data implies the possibility of the model being applied to precision feeding at the farm level without dedicated devices. However, this system still requires a high level of computing resources on a large scale. If slaughterhouses set up a server for the region, farms could identify individual cattle with no computer resources, and slaughterhouses could further strengthen the system for grading. Expanding the scope, data from cattle, including genetic traits, can be easily accumulated on a national scale.

Deep learning is a very convenient technology used to reduce human resources including labor and concentration because the computer trains using the traits of image data without human supervision. With the current technology, while deep learning offers convenience in automating the training process, it remains challenging to determine the key traits for accurately classifying cattle in images [30]. The inner workings of the model and how it achieves classification are not fully understood. However, the development of an accurate classification model for cattle individual based on their muzzle patterns, as demonstrated in this study, holds significant meaning. It suggests the potential for non-contact individual identification through image analysis. Future studies will focus on reducing the complexity of the main data to eliminate unnecessary variables and identify the key traits for classification, as well as trying to make generalized models for various species. 

## 5. Conclusions

This study aimed to develop a deep learning model based on muzzle images, with a focus on *Hanwoo* cattle. The small Efficientnet V2 model presented a high level of predictive accuracy. While showing subtly better performance compared with same model learned from *Holstein,* the performance of the *Hanwoo* model implies the potential for real-farm usage in terms of precision feeding and management strategies, as well as facilitating traceability on a national scale. Furthermore, the proposed approach introduces this perspective by utilizing inherent physical characteristics of cattle for identification purposes, reducing the reliance on ear tags. 

As the field of deep learning continues to evolve with developments in computer hardware and the tendency for such hardware to become cheaper, further research could investigate refining the accuracy of the model, exploring other traits and extending its applicability. Improved studies will offer improved precision breeding and more efficient management, bringing advantages to both farmers and the industry.

## Figures and Tables

**Figure 1 animals-13-02856-f001:**
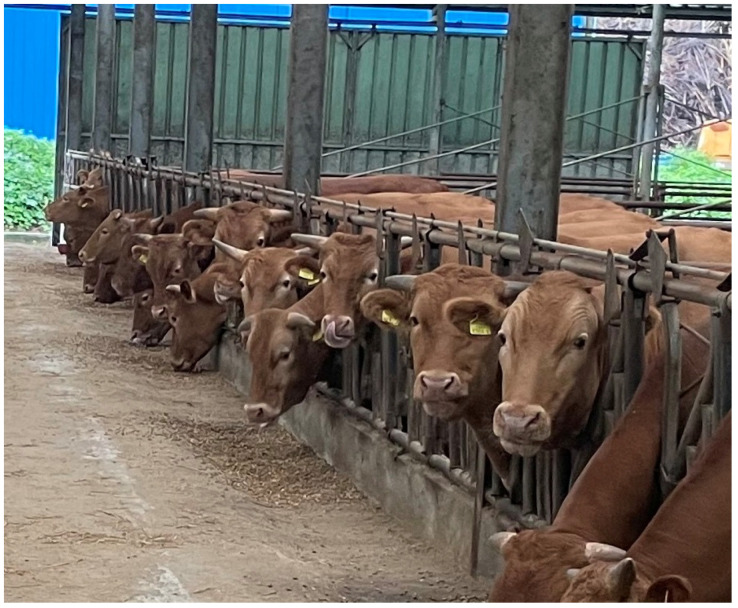
Resemblance of the appearance among *Hanwoo* cattle.

**Figure 2 animals-13-02856-f002:**
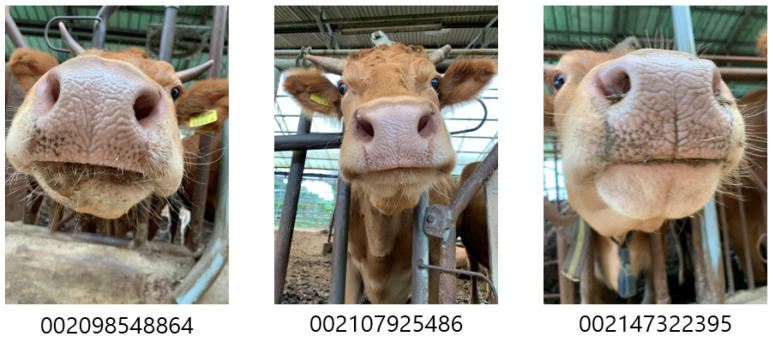
Samples of image data with national livestock traceability numbers. The national livestock traceability number is a unique identifier assigned to cattle at birth by the nation in order to identify individuals within the cattle population.

**Figure 3 animals-13-02856-f003:**
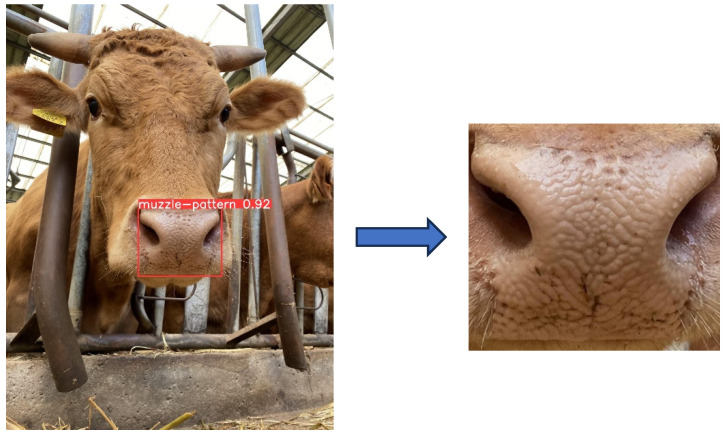
Example of cropping muzzle in the image with YOLO v8-based crop model.

**Figure 4 animals-13-02856-f004:**
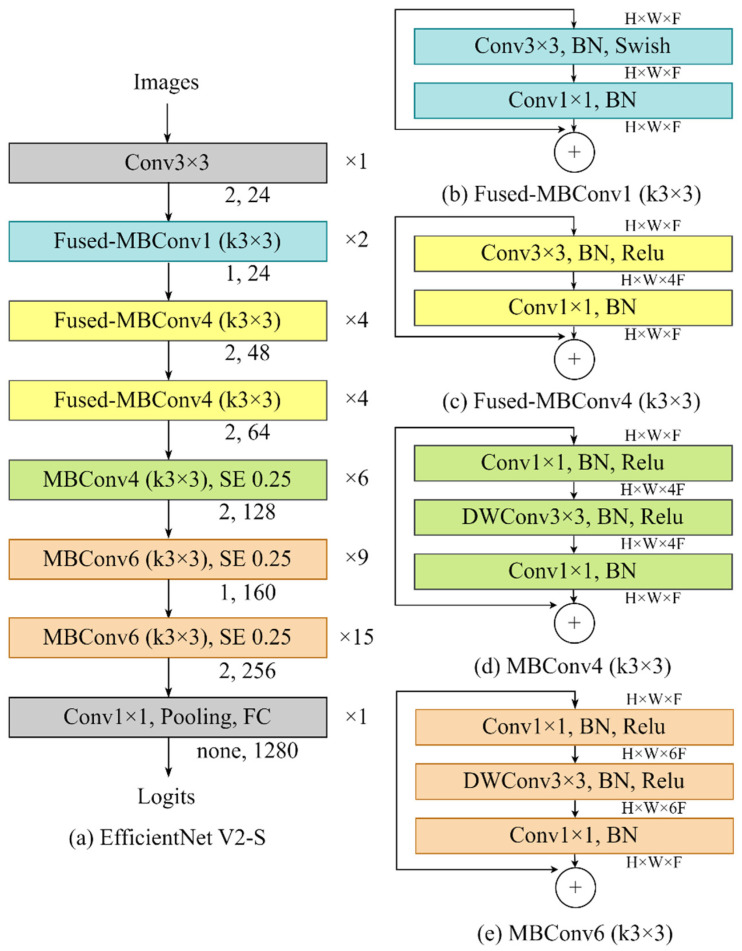
Efficientnet v2-s Architecture—(**a**) represents the schematic of the model from image input to logit output. The numbers indicated on the right side of the boxes represent the layer numbers, while the numbers displayed below the boxes indicate the corresponding stride and channels. (**b**–**e**) are the schematic of each specific algorithm in (**a**); Fused-MBConv1, Fused-MBConv4 (k3 × 3), MBConv4 (k3 × 3), and MBConv6 (k3 × 3), respectively. The abbreviations are expanded as follows: MBConv as mobile vision convolutional network, SE as squeeze and excitation, FC as full connect, DW as depth wise, BN as batch normalization, H as height, W as width and F as number of channels.

**Figure 5 animals-13-02856-f005:**
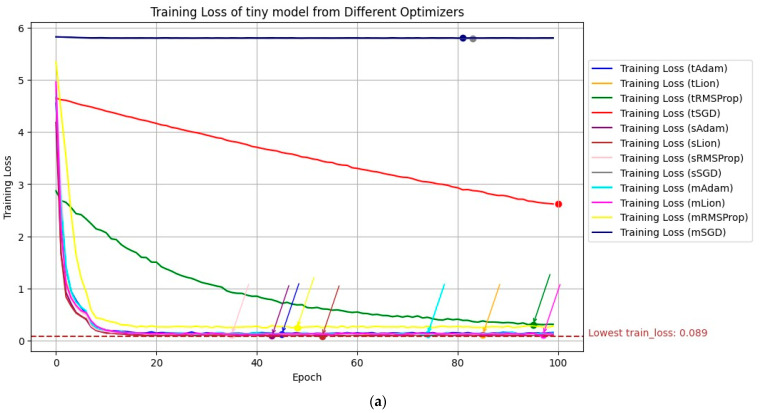
Training loss (**a**) and training accuracy (**b**) visualized through transfer learning across all models.

**Figure 6 animals-13-02856-f006:**
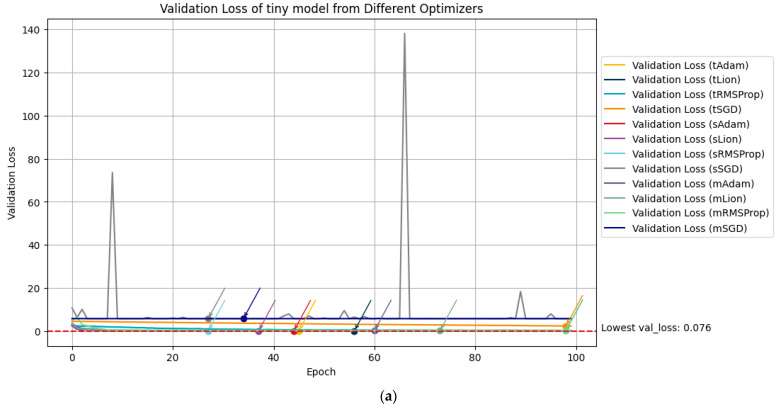
Validation loss (**a**) and validation accuracy (**b**) visualized through transfer learning across all models.

**Table 1 animals-13-02856-t001:** Location, number of *Hanwoo* cattle and number of images enrolled in dataset.

Region	Location (Latitude, Longitude)	Number of Animals	Images
Jeongeup-si	35.62934, 126.87748	235	6160
Wonju-si	37.20481, 127.5141	101	3070
Sum		336	

**Table 2 animals-13-02856-t002:** Parameters used through data transformation for train, validation and test dataset.

Parameters	Train Data	Validation Data	Test Data
Resize	224 × 224 pixels	224 × 224 pixels	224 × 224 pixels
Horizontal flip	Random	Random	None
Vertical flip	Random	Random	None
Rotation	0–20 degrees	0–20 degrees	None
RGB mean value RGB standard deviation	[0.485, 0.456, 0.406], [0.229, 0.224, 0.225]	[0.485, 0.456, 0.406], [0.229, 0.224, 0.225]	[0.485, 0.456, 0.406], [0.229, 0.224, 0.225]

**Table 3 animals-13-02856-t003:** Concept of two-by-two confusion matrix as classification interpretation.

	Predicted Negative	Predicted Positive
Actual negative	True negative	False positive
Actual positive	False negative	True positive

**Table 4 animals-13-02856-t004:** Comparison of Efficientnet v2 models in detail of parameters, Giga Multiply Accumulates (GMACs) and number of activations.

	Efficientnet v2 Tiny	Efficientnet v2 Small	Efficientnet v2 Medium
Parameters (million)	13.6	23.9	53.2
GMACs (giga)	1.9	4.9	12.7
Activation (million)	9.9	21.4	47.1

**Table 5 animals-13-02856-t005:** Training loss, training accuracy, validation loss and validation accuracy derived from the epoch with the highest validation accuracy achieved through transfer learning across all models.

Model	Best Epoch	Epoch Time (s)	Train_loss	Train_acc	Val_loss	Val_acc
Tiny-SGD	97	59	2.638	0.489	2.381	0.539
Tiny-RMSprop	97	62	0.310	0.917	0.261	0.932
Tiny-Adam	44	58.5	0.133	0.968	0.104	0.976
Tiny-Lion	55	54.9	0.136	0.966	0.098	0.976
Small-SGD	38	62.2	5.791	0.008	5.793	0.014
Small-RMSprop	26	59.1	0.141	0.962	0.091	0.977
Small-Adam	67	58.5	0.111	0.972	0.090	0.978
Small-Lion	36	59	0.090	0.976	0.077	0.981
Medium-SGD	89	60.5	5.803	0.007	5.804	0.011
Medium-RMSprop	97	60.2	0.265	0.937	0.371	0.907
Medium-Adam	51	65.3	0.138	0.968	0.231	0.944
Medium-Lion	22	65.1	0.131	0.968	0.256	0.942

Train_loss—training loss; Train_acc—training accuracy; Val_loss—vValidation loss; Val_acc—validation accuracy.

**Table 6 animals-13-02856-t006:** Accuracy of prediction from top five models on validation accuracy and epoch time on test data with time elapsed with numbers of entire errors and multiple error prediction on same subject (repeated error).

Model	Test Accuracy	Testing Time (s)	Error	Repeated Error
Tiny-Adam	0.967	18.6	92	14
Tiny-Lion	0.967	18.2	92	12
Small-RMSprop	0.968	18.3	91	3
Small-Adam	0.970	17.9	95	9
Small-Lion	0.965	18.2	100	18

## Data Availability

Data sharing is not applicable to this article.

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
