# Peer review of "Identification of Individual Hanwoo Cattle by Muzzle Pattern Images through Deep Learning"

_animals, 2023, doi:10.3390/ani13182856_

Round 1

Reviewer 1 Report (Previous Reviewer 1)

Dear authors 

Thank you for welcoming my requests of change. I believe that the manuscript improved greatly and can be accepted for publication. 

Author Response

Reviewer 1

Comments and suggestions for authors

Dear authors 

Thank you for welcoming my requests of change. I believe that the manuscript improved greatly and can be accepted for publication. 

Answer: Thank you for the feedback on this paper. The manuscript has been improved during the revision owing to your feedback.

Reviewer 2 Report (Previous Reviewer 2)

The detailed comments and suggestions are in the attached document.

Some syntax errors have been found and indicated in the text. The specified sentences should be refined.

Author Response

Reviewer 2

Comments and suggestions for authors

The detailed comments and suggestions are in the attached document.

  1. Lines 20-21: It should not include too many training details in the Abstract section.

Answer: The sentence has been deleted.

  1. Lines 45-46: It seems not make sense that resemblance of the appearance among Hanwoo cattle might increase the risk of tag loss.

Answer: The intention I was trying to convey was that if two or more Hanwoo cattle had lost tags simultaneously, they have no identification key so it would be difficult to identify individuals. So the sentence has changed to ‘Fosgate suggested relying solely on ear tag for long-term identification might not be sufficient and resemblance of the appearance among Hanwoo cattle might increase the risk of failure in individual recognition due to tag loss’ in lines 46-49 in the new paper.

  1. Lines 46–47: This expression is quite questionable. To my knowledge, ear tags can achieve almost 100% identification accuracy for a very long period of time, while it seems that identification using deep learning can't make it, at least so from your experimental results.

Answer: I fully agree with your feedback. The sentence has been changed to ‘To address the issue of ear-tag loss, employing supplementary identification methods can aid in ensuring recognition even in cases of ear-tag loss’ in line 49-51 in the new paper.

  1. Line 49: Syntax error

Answer: ‘Is’ after ear-tag has been changed to ‘are’ in line 52 in the new paper.

  1. Lines 64-68: The authors should cite some published papers to justify these views.

Answer: Study of Jia deng, ‘What Does Classifying More Than 10,000 Image Categories Tell Us?’ was cited in the new paper.

  1. Line 70: Syntax error here! What's more, the view the authors want to express in this sentence is also doubtful. The annotation of training data, the training and the construction of deep learning models require human supervision, to a large extent.

Answer: The mentioned lines were rephrased ‘If machine learning algorithms provide inaccurate predictions, engineers need to intervene and make adjustments. When using deep learning models, the algorithm can assess the accuracy of predictions through its own neural network, eliminating the need for hu-man assistance.’, to avoid ambiguity in meaning in lines 73-76 in the new paper.

  1. Lines 82: The resolution of the cattle image in Figure 2 is a little low and not the same as the image in Figure 3.

Answer: The image in Figure 2 has been replaced with a higher-resolution image in line 91 in the new paper.

8. Lines 84~85: Syntax error here. There are some syntax errors in the paper, especially in the usage of the definite article "the". I just point out some evident, the authors should carefully check and correct this problem throughout the text before submitting again.

Answer: ‘National livestock traceability number, unique number of cattle given from birth by nation to identify within cattle’ was changed to ‘The national livestock traceability number is a unique identifier assigned to cattle at birth by the nation to identify individuals within the cattle population.’ Also, checked syntax errors again throughout the text in lines 93-95 in the new paper. Also, syntax errors were checked and changed during revision. Changed parts are addressed at the end of this response letter.

  1. Line 95: The inconsistent capitalization of the first letter of the first word in a sentence.

Answer: ‘1. configure the endline setting to include whole muzzle area’ was changed to ‘1. Configure the endline to include whole muzzle area’ for the consistent capitalization in line 106 in the new paper.

  1. Line 119: It should be changed to the mean value and standard deviation.

Answer: ‘RGB’ was changed to ‘RGB mean value’ and ‘Normalization’ was changed to ‘RGB standard deviation’ in line 129 in the new paper.

  1. Line 158: The authors should give a figure that fully describes the architecture of Efficientnet v2.

Answer: The architecture of Efficientnet is described in lines 165-191, including a figure in the new paper.

  1. Lines 239-240: The table and its caption should not be displayed across pages.

Answer: Caption and the table 5 are now displayed on the same page in lines 270-272 in the new paper.

  1. Line 254: The authors should include the limitations of their study in this section.

Answer: Limitations of the study are added in lines 301-316 in the new paper.

  1. Lines 255-256: It is not recommended to cite some content in unpublished research (especially some not publicly available).

Answer: Citation has been deleted in the new paper in line 286 in the new paper.

  1. Lines 283-286: It is better to try another way of expression without referring to something in the prequel's unpublished research.

Answer: Instead of referring to unpublished research, entire paragraph was changed in lines 329-342 to also address the next feedback you gave.

  1. Lines 286-289: It is quite questionable. The model gets overfitted not because of the diversity of the muzzle patterns that makes the model cannot learn the distinctive patterns (on the contrary it benefits the model's generalization to new muzzle patterns identification), but because it learns some noise muzzle patterns.

Answer: During the revision of the paragraph, the notion that while the model's capacity to learn various traits can contribute to enhancing accuracy, it might pose challenges when dealing with diverse breeds and larger datasets in the future was included in lines 329-342.

  1. Lines 314-316: The future study should be closely related to the limitations (not mentioned above) of the study.

Answer: Taking your feedback for the first version of this paper into consideration, the additional limitation of creating a generalized model for various species were added in line 367 in the new paper.

Comments on the Quality of English Language

Some syntax errors have been found and indicated in the text. The specified sentences should be refined.

Answer: Including the syntax error occurred you have mentioned in the attached document, entire manuscript has been double-checked about syntax errors during revision.

Reviewer 3 Report (Previous Reviewer 3)

First of all, I wanted to congratulate the authors for presenting this significant research project. Animal identification via artificial intelligence and/or data-based models has been a challenge while researchers are investigating alternative identification methods for RFID systems. This work is timely and informative to the precision livestock farming sector.

All of my comments on the previous version of the manuscript were addressed properly. Notably, the recently added muzzle (object) detection model and the comparison of different neural networks elevated the manuscript. Given the improvement, I would suggest this manuscript proceed to publication.

Please double-check text rendering and document formatting. For instance, the index for Section 4 was noted as a superscript.

Author Response

Reviewer 3

Comments and suggestions for authors

First of all, I wanted to congratulate the authors for presenting this significant research project. Animal identification via artificial intelligence and/or data-based models has been a challenge while researchers are investigating alternative identification methods for RFID systems. This work is timely and informative to the precision livestock farming sector.

All of my comments on the previous version of the manuscript were addressed properly. Notably, the recently added muzzle (object) detection model and the comparison of different neural networks elevated the manuscript. Given the improvement, I would suggest this manuscript proceed to publication.

Answer: Thank you for the feedback on this paper. The manuscript has been improved during the revision owing to your feedback.

Comments on the Quality of English Language

Please double-check text rendering and document formatting. For instance, the index for Section 4 was noted as a superscript.

Answer: Including the change of index for section 4, text rendering and document formatting are double-checked during revision and highlighted in the new paper.

Round 2

Reviewer 2 Report (Previous Reviewer 2)

This manuscript improved greatly and can be accepted for publication.

Minor editing of English language required.

This manuscript is a resubmission of an earlier submission. The following is a list of the peer review reports and author responses from that submission.

Round 1

Reviewer 1 Report

Dear authors,

Many thanks for this piece of work, which I read with interest. I must admit that I found difficult agreeing with your rationale. I explain why. The muzzle image doesn't represent a bioeconomic morphological trait,  like for instance the morphological aspect that predict heritable traits of production yields. The muzzle print is true instead to identify animals on the basis of a biological traits like the finger print for humans, but not the entire muzzle per se. Howòever, as an external part of the animal body, highly exposed to the environmental insults,  if traumatic event or scars occur. And this was abandoned because technology took over. Being always wet it can get covered by chips or debris that rendered automatic animal recognition in the barn somewhat impaired in routine field activities. Finally, animal identification  really stepped forward in the last decades, and all individuals are tagged with devices base on RFID technology. And also farm equipments work on the recognition of each single animal thanks to RFID (Cappai et al., 2014 Small Rum Res. 117(2-3):169-175 doi:10.1016/j.smallrumres.2013.12.031; Cappai et al., 2018 COMPAG, 144:324-328  doi:10.1016/j.compag.2017.11.002) to dispense feed for precision feeding aspects or at milking when in cows are in the parlour. Moreover, mandatory methods of animal identification are different from that you propose here.

Thus, based on such aspects, I would ask authors to contextualize the scope and advancement of the state of the art, also in the light and perspectives of their findings directly applicable into practice, because I missed those points. So please, add such background to improve the quality of the experimental design and describe a rationale behind this activity.

Thank you. 

Author Response

Reviewer 1

Comments and suggestions for authors

Many thanks for this piece of work, which I read with interest. I must admit that I found difficult agreeing with your rationale. I explain why. The muzzle image doesn't represent a bioeconomic morphological trait, like for instance the morphological aspect that predict heritable traits of production yields. The muzzle print is true instead to identify animals on the basis of a biological traits like the finger print for humans, but not the entire muzzle per se. However, as an external part of the animal body, highly exposed to the environmental insults, if traumatic event or scars occur. And this was abandoned because technology took over. Being always wet it can get covered by chips or debris that rendered automatic animal recognition in the barn somewhat impaired in routine field activities. Finally, animal identification really stepped forward in the last decades, and all individuals are tagged with devices base on RFID technology. And also farm equipments work on the recognition of each single animal thanks to RFID (Cappai et al., 2014 Small Rum Res. 117(2-3):169-175 doi:10.1016/j.smallrumres.2013.12.031; Cappai et al., 2018 COMPAG, 144:324-328  doi:10.1016/j.compag.2017.11.002) to dispense feed for precision feeding aspects or at milking when in cows are in the parlour. Moreover, mandatory methods of animal identification are different from that you propose here.

Thus, based on such aspects, I would ask authors to contextualize the scope and advancement of the state of the art, also in the light and perspectives of their findings directly applicable into practice, because I missed those points. So please, add such background to improve the quality of the experimental design and describe a rationale behind this activity.

1. Comment: The muzzle image doesn't represent a bioeconomic morphological trait, like for instance the morphological aspect that predict heritable traits of production yields.

Answer: The intention I was trying to convey was that genetic traits could be improved without error of information through precise entity recognition but deleted it due to ambiguity

2. Comment: The muzzle print is true instead to identify animals on the basis of a biological traits like the fingerprint for humans, but not the entire muzzle per se.

Answer: In the new version of paper, muzzle area of the image was cropped for training the model.

3. Comment: However, as an external part of the animal body, highly exposed to the environmental insults, if traumatic event or scars occur.

Answer: While capturing images from 646 cows, I did not come across the mentioned cases. However, I agree with your opinion, and cases in your comment should be definitely concerned. So the flow of the paper was changed from “Muzzle pattern can replace the ear tag” to “Muzzle pattern can subsidize the ear tag”.

4. Comment: Being always wet it can get covered by chips or debris that rendered automatic animal recognition in the barn somewhat impaired in routine field activities.

Answer: During preparing the dataset, I have visited five farms in various time slots. But in all cases, it was always licked to stay tidy besides about 30 minutes after feeding time.

5. Comment: Finally, animal identification really stepped forward in the last decades, and all individuals are tagged with devices base on RFID technology.

Answer: I agree with the perfect accuracy and convenience of RFID technology. But RFID chips are occasionally detached according to the study mentioned on line 44 in the new version. If muzzle pattern is not altered by trauma, methods of recognizing individual animals through parts of their cattle body could subsidize the RFID tag methods.

6. Comment: Thus, based on such aspects, I would ask authors to contextualize the scope and advancement of the state of the art, also in the light and perspectives of their findings directly applicable into practice, because I missed those points.

Answer: In the result of paper, highest accuracy of the model was up to 98 percent. It is quite precise, but not enough to apply directly into practice. But as the field of artificial intelligence and computer hardware are evolving fast, using images to identify cattle can be practical in the near future, and I mentioned this on line 320-323 in the new version of paper.

7. Comment: So please, add such background to improve the quality of the experimental design and describe a rationale behind this activity.

Answer: Thanks to your comments, I have added background in introduction section to describe a rationale of this study and tried to improve the quality of experimental design.

Reviewer 2 Report

It is very interesting to perform cattle identification by muzzle patterns. However, there are a lot of problems in your paper, such as the deep learning model used in the study is too out-of-date and lacks the improvement on it. Apart from that, there are a lot of other problems needed to be improved, such as grammar errors and unprofessional expressions in your paper. Detailed comments and recommendations are in the attached document. 

There are many grammatical and syntax errors in your paper, and the writing is also not very professional and academic, see the attached document above for details. 

Author Response

Reviewer 2

Comments and suggestions for authors

It is very interesting to perform cattle identification by muzzle patterns. However, there are a lot of problems in your paper, such as the deep learning model used in the study is too out-of-date and lacks the improvement on it. Apart from that, there are a lot of other problems needed to be improved, such as grammar errors and unprofessional expressions in your paper. Detailed comments and recommendations are in the attached document.

Answer: Deep learning model used in the study was changed ResNet to Efficientnet V2 (2019). It has been several years from the release, but this model is still showing the best performance for image classification. Also, trending optimizers were added: RMSProp, Adam and Lion (2023). Grammar errors and expressions are examined through the revision.

  1. Line 10: Syntax error

Answer: The sentence has changed by deleting “differ by each individual” in line 10.

  1. Lines 22-23: Preferably, PyTorch is a framework rather than a library.

Answer: The sentence was deleted during the revision.

  1. Lines 37–38: Can the identification improve genetic traits?

Answer: The intention I was trying to convey is that genetic traits could be improved without error of information through precise entity recognition but deleted it due to ambiguity.

  1. Lines 50-51: Syntax error

Answer: Changed sentence to ‘Several studies have been conducted to use muzzle patterns to identify cattle and have provided results with various species of cattle.’ In line 71-72.

  1. Line 55: Do more images improve processing speed?

Answer: Upon your feedback, the mentioned phrase was deleted, and entire paragraph was revised in line 64-70.

  1. Lines 66–68: This is not necessary. This is an academic paper, not a report to the general public without relevant professional background.

Answer: The mentioned lines were deleted upon your feedback. Thank you.

  1. Lines 80: it is obvious that not every cattle is taken with at least 20 images.

Answer: During the revision of paper, data for Holstein has been deleted because focus of paper had changed to only Hanwoo. Also, line 84-86 in previous version ‘At least 20 pictures were taken per individual. To avoid overfitting the model, five pictures were taken from four each different times and some images with foreign substances were added.’ was changed to ‘. For each individual cattle, approximately 20 images were captured, and five pictures were taken from four each different times to avoid overfitting the model and some images with foreign substances were added.’ in line 90-92 in the new version.

  1. Line 96: What was the batch size set to?

Answer: The number of batch size was 32 and presented on line 127-128 in the new paper.

  1. Line 110: It should include the data augmentation you used in the model training and validation.
    Answer: Specific transformations used during the model training and validations were added to Table 2 in line 119, and also described in line 112-116.

  1. Line 131: SGD is used to update parameters in the model rather than for the calculation of loss

Answer: Thanks to your feedback, I reexamined SGD and rewrite the section in line 179-185.

  1. Line 138: syntax error

Answer: The sentence has been deleted because contents describing SGD was rewritten during revision.

  1. Line 142: unprofessional expression

Answer: The sentence has been deleted because contents describing SGD was rewritten during revision.

  1. Line 176: misspelling here?

Answer: Sorry for the mistake. The result was changed during the revision and changed the name of the result.

  1. Lines 225: Resnet is an old classification network (proposed in 2015), the authors should include more comparison results with other up-to-date models.

Answer: Thank you for the feedback. During the revision, used model ResNet was changed to Efficientnet V2 with three volumes (2019). Efficientnet models are opted because it currently holds the highest performance in the field of image classification. Also, trending optimizers were added too: RMSProp, Adam and Lion (2023).

  1. Lines 310-311: Why not train the model with a combined dataset of Havwoo cattle and Holstein cattle to boost the model's generalization ability? and ensure a more in-depth knowledge of previously completed studies on the subject matter.

Answer: Model in the previous version was trained separately in order to compare the learning process of two different species. But in new version, only the Hanwoo classifying model is presented, while Holstein model is briefly mentioned only in the discussion section in line 276-289.

  1. Shouldn't this sentence be removed?

Answer: Thanks to your feedback, the sentence was deleted.

Reviewer 3 Report

Please find and refer to the comments below for the overall recommendation.

1) This work did not develop a novel deep learning model. Instead, a ResNet-based transfer learning was adapted for the cattle ID classification problem. Further, other advanced deep learning architectures e.g., transformer and Inception were not investigated or justified for absence in this study.

2) There are existing studies that utilized deep learning techniques for cattle ID using muzzle patterns. For instance:

Kumar, Santosh, et al. "Deep learning framework for recognition of cattle using muzzle point image pattern." Measurement 116 (2018): 1-17.

In addition, references to this manuscript are a bit dated. More recent works from 2020-23 can be included in this study.

In summary, there is a lack of significance and novelty in this study.

3) No instruction, guideline, or object detection model for muzzle area was provided for cropping muzzle images. Lines 284-286: although a high classification accuracy may be achieved, the experiment cannot be reproduced.

4) Table 6 is beneficial to summarize results and/or prediction accuracy, but does not help with model diagnostics with misclassified IDs.

5) Lines 291-293: Not really. Here only the convenience of existing deep learning architectures is reflected. However, the data aspect will not be the case for reducing human resources. Annotation and data formatting take significant amounts of time whenever a new cattle individual is introduced, whereas the deep learning model needs to be retrained and the new ID needs to be annotated.

6) Lines 310-311: ResNet-101 yielded highest accuracy only because the ResNet models were investigated in this work. This does not necessitate the conclusion statement.

7) The major gap in animal identification is still not addressed in this study: in cattle cohorts, animals are frequently in and out. Machine learning or deep learning algorithms may work well for close-set classification problems, but not really for open-set animal identifications. 

Author Response

Reviewer 3

Comments and suggestions for authors

Please find and refer to the comments below for the overall recommendation.

1. Question: This work did not develop a novel deep learning model. Instead, a ResNet-based transfer learning was adapted for the cattle ID classification problem. Further, other advanced deep learning architectures e.g., transformer and Inception were not investigated or justified for absence in this study.

Answer: The terms of ‘developing deep learning model’ in the paper were changed to ‘deep learning model learned through transfer learning’. Also, data transformation was added to Table 2 in line 119, with additional description in line 112-116

2. Question: There are existing studies that utilized deep learning techniques for cattle ID using muzzle patterns. For instance:

Kumar, Santosh, et al. "Deep learning framework for recognition of cattle using muzzle point image pattern." Measurement 116 (2018): 1-17.

In addition, references to this manuscript are a bit dated. More recent works from 2020-23 can be included in this study.

In summary, there is a lack of significance and novelty in this study.

Answer: There are similar studies about utilizing deep learning techniques for identifications, but there was no study for Hanwoo cattle, which have very similar color in monotone in the muzzle area. Cattle species used for the studies had various appearance. Because image classification is based on pixel traits in the image, I believe that new research is required to apply. These contents were mentioned in line 44-46 in the new version of paper. Also, references of recent works were added too.

3. Question: No instruction, guideline, or object detection model for muzzle area was provided for cropping muzzle images. Lines 284-286: although a high classification accuracy may be achieved, the experiment cannot be reproduced.

Answer: Instruction, guideline for cropping muzzle images were added in line 93-105.

4. Question: Table 6 is beneficial to summarize results and/or prediction accuracy, but does not help with model diagnostics with misclassified IDs

Answer: Table 6 has been moved to Table 3 in line 142, to describe the accuracy of the model and deleted the contents about model diagnostics with misclassified IDs.

5. Question: Lines 291-293: Not really. Here only the convenience of existing deep learning architectures is reflected. However, the data aspect will not be the case for reducing human resources. Annotation and data formatting take significant amounts of time whenever a new cattle individual is introduced, whereas the deep learning model needs to be retrained and the new ID needs to be annotated.

Answer: I agree with your feedback about the time-taking process for annotating the new data. So, the content about auto-crop model for the muzzle pattern area was added in line 93-100.

6. Question: Lines 310-311: ResNet-101 yielded highest accuracy only because the ResNet models were investigated in this work. This does not necessitate the conclusion statement.

Answer: In the new version of paper, several models with optimizers were tested to outbound images not learned through and accuracy of prediction on new data is presented in the result and discussion part. The conclusion of the study has changed through the revision of the paper.

7. Question: The major gap in animal identification is still not addressed in this study: in cattle cohorts, animals are frequently in and out. Machine learning or deep learning algorithms may work well for close-set classification problems, but not really for open-set animal identifications. 

Answer: In countries with small husbandry lands, including my country Korea, are raising cattle in small facility and they are not abled to go out from the facility. Also, if there is subsidizing identification method, double check will enhance the reliability of data in case of the grading assessments in slaughterhouse for example, mentioned in line 49-52.

Others

Title changes in line 2
Identification of Hanwoo and Holstein Friesian individual by muzzle pattern using deep learning algorithm
-> Identification of individual Hanwoo cattle by the images of muzzle pattern through deep learning

line 10
Cattle muzzle has unique patterns differ by each individual, which can be used as a biometric identification key.
-> Cattle muzzle has unique patterns, which can be used as a biometric identification key.

table 1 deleted in line 320-321
As the field of deep learning continues to evolve, further research can investigate into refining the model accuracy, exploring other traits and extend its applicability.
->As the field of deep learning continues to evolve with developments of computer hardware and tendency becoming cheaper, further research can investigate into refining the model accuracy, exploring other traits and extend its applicability.developement of computer hardware and tendency becoming cheaper